

# IRimage: open source software for processing images from infrared thermal cameras

Gustavo Pereyra Irujo

Instituto Nacional de Tecnología Agropecuaria (INTA), Consejo Nacional de Investigaciones Científicas y Técnicas (CONICET), Balcarce, Buenos Aires, Argentina

## ABSTRACT

IRimage aims at increasing throughput, accuracy and reproducibility of results obtained from thermal images, especially those produced with affordable, consumer-oriented cameras. IRimage processes thermal images, extracting raw data and calculating temperature values with an open and fully documented algorithm, making this data available for further processing using image analysis software. It also allows the making of reproducible measurements of the temperature of objects in a series of images, and produce visual outputs (images and videos) suitable for scientific reporting. IRimage is implemented in a scripting language of the scientific image analysis software ImageJ, allowing its use through a graphical user interface and also allowing for an easy modification or expansion of its functionality. IRimage's results were consistent with those of standard software for 15 camera models of the most widely used brand. An example use case is also presented, in which IRimage was used to efficiently process hundreds of thermal images to reveal subtle differences in the daily pattern of leaf temperature of plants subjected to different soil water contents. IRimage's functionalities make it better suited for research purposes than many currently available alternatives, and could contribute to making affordable consumer-grade thermal cameras useful for reproducible research.

## INTRODUCTION

Thermal imaging has many uses in biological, medical and environmental research (*Kastberger & Stachl, 2003*). In recent years, thermal cameras have lowered their price, and affordable consumer cameras are now available for as little as 300USD, either as stand-alone devices or smartphone attachments (*Haglund & Schönborn, 2019*). These cameras, in spite of being marketed as consumer devices, have been proven to be suitable as scientific instruments for research (*Pereyra Irujo et al., 2015*; *Klaessens, Veen & Verdaasdonk, 2017*; *Razani, Parkhimchyk & Tabatabaei, 2018*; *Petrie et al., 2019*; *Van Doremalen et al., 2019*; *Nosrati et al., 2020*) and have the potential to greatly improve access to thermography in many scientific fields, especially for budget-limited scientists.

Thermal cameras do not measure temperature directly. Temperature is estimated from measured infrared radiation captured by the sensor in the camera, through a series of

Corresponding author
Gustavo Pereyra Irujo,
pereyrairujo.gustavo@conicet.gov.ar

equations and using a set of parameters, some provided by the user through the camera interface, and others which are set during calibration. The software coupled to these cameras is usually closed-source, which does not allow the user to know the exact algorithms used to obtain the temperature measurements and the final image. For a thermal camera (or any sensor) to be useful for research, the user should be able to have control over (or at least information about) the processing steps between the raw sensor data and the final measurement (*Dryden et al., 2017*).

Software provided with low-cost infrared cameras, besides being closed-source, has usually limited functionality, since it is aimed at non-scientific users. This kind of software only allows for temperature measurements of manually selected points or areas in the image (*e.g.*, *Nosrati et al., 2020*), which is impractical with a large quantity of images, and hinders reproducibility of results. Parameters used for temperature calculations, also required for reproducibility, are generally under-reported (*Harrap et al., 2018*), a problem which can be made worse with consumer-oriented analysis software (especially smartphone apps) that hide the real values of parameters behind simple user options. Besides obtaining and analyzing temperature data in numerical form, false-color images representing the temperature values are usually necessary for visualizing and reporting. Similarly to scientific plots of data, it is important that this representation of temperature data is quantitatively accurate. In spite of this, some of the default color palettes used in consumer-oriented software are selected for aesthetic reasons and do not meet the necessary criteria for scientific reporting (*Crameri, Shephard & Heron, 2020*).

IRimage was developed with the aim of overcoming those problems and allowing researchers to increase throughput, accuracy and reproducibility of results obtained from thermal images, especially those produced with affordable, consumer-oriented cameras. It allows researchers to extract raw data from thermal images and calculate temperature values with an open and fully documented algorithm, making this data available for further processing using standard image analysis or statistical software. It also allows the making of reproducible measurements of the temperature of objects in image sequences, along with other outputs that are useful for further analysis and reporting, such as image timestamps, parameters used for temperature estimations, and customized false-color images and videos that use a scientifically accurate color palette. IRimage was initially developed as an in-house simple tool which was used to benchmark a low-cost thermal camera (the "FLIR One" smartphone attachment, *Pereyra Irujo et al., 2015*) and to analyse thermal images of wheat varieties (*Cacciabue, 2016*), and was later developed further in order to make it suitable for a wider range of scientific applications. In this article, IRimage implementation and usage is described, along with its comparison against standard software, and an example use case that highlights the utility of some of its functions.

## MATERIALS & METHODS

### Theoretical background for temperature calculations

One of the main objectives of IRimage is to provide an implementation of an algorithm which is open not only from a software point of view (as in the definition of open source

software), but also open in the sense of being transparent and understandable to the end user, and thus available for scientific scrutiny, customization or extension. To this end, a detailed explanation of the theoretical background of the algorithm used in IRimage is presented here.

### Relationship between temperature and infrared radiation

Thermal cameras are based on the detection of infrared radiation emitted from objects by means of an array of sensors. Each of these sensors generates a digital signal ($S$), which is a function of radiance ($L$). Radiance is the radiant flux (*i.e.,* amount of energy emitted, reflected, transmitted or received per unit time, usually measured in Watts, W) per unit surface and solid angle (in W sr$^{-1}$ m$^{-2}$). The relationship between the signal ($S$) resulting from the voltage/current generated by the sensor and the associated electronics (usually quantified as Digital Numbers; DN) and $L$ is usually linear, and gain ($G$) and offset ($O$) factors can be calibrated:

$$S = G \cdot L + O. \tag{1}$$

Measuring radiance can be used to estimate temperature because the total amount of energy emitted by an object is a function of absolute temperature to the fourth power (according to the Stefan–Boltzmann law). The emission is, however, not equal at different wavelengths (even for a perfect emitter, *i.e.,* a black body): according to Wien's displacement law, the wavelength corresponding to the peak of emission also depends on temperature. For instance, the peak emission of the sun is around 500nm (in the visible portion of the spectrum), while that of a body at 25 °C is around 10μm (in the far infrared). Since detectors are only sensitive to part of the spectrum, it is necessary to take into account only the spectral radiance ($L_\lambda$) for a given wavelength (according to the Lambert's cosine law and the Planck's law) which is equal to:

$$L_\lambda = \varepsilon \cdot \frac{2hc^2}{\lambda^5} \cdot \frac{1}{e^{\frac{hc}{\lambda kT}} - 1}, \tag{2}$$

where $\varepsilon$ is the emissivity of the surface, $h$ is the Planck constant, $c$ is the speed of light in the medium, $\lambda$ is the wavelength, $k$ is the Boltzmann constant, and $T$ is the absolute temperature of that surface (in kelvins). This equation needs to be integrated over the spectral band corresponding to the detector sensitivity (short-wavelength: 1.4–3 μm, mid-wavelength: 3–8 μm, or long-wavelength: 8–15 μm, depending on the type of sensor) or, for simplicity, be multiplied by the spectral sensitivity range (*Gaussorgues, 1994*). For a given camera (*i.e.,* combination of electronics, sensors and lenses) this equation can be simplified as:

$$L_\lambda = \varepsilon \cdot \frac{1}{R \cdot (e^{\frac{B}{T}} - 1)}, \tag{3}$$

where $B$ and $R$ are camera calibration parameters (together with $G$ and $O$). In some cases, the constant 1 is also stored as calibration parameter $F$.

By combining Eqs. (1) and (3), it is possible to obtain an equation that represents the relationship between $S$ and $T$ for a given sensor, and can be used for its calibration:

$$S = G \cdot \varepsilon \cdot \frac{1}{R \cdot (e^{\frac{B}{T}} - 1)} + O. \tag{4}$$

### Sources of radiation

The radiation received by the camera sensor is not equal to the radiation emitted by the object(s) in its field of view. Depending on the emissivity of the object's surface, radiation reflected by the object's surface can contribute significantly to the radiation received by the sensor. Furthermore, this radiation is then attenuated by the atmosphere (mainly by water molecules) even at short distances (*Minkina & Klecha, 2016*). Taking this into account, the signal detected by the sensor ($S$) can be considered to be composed of three terms:

$$S = \tau \cdot S_{obj} + \tau \cdot S_{refl} + S_{atm}, \tag{5}$$

where the first term is the equivalent digital signal originating from the target object ($S_{obj}$), attenuated by the atmosphere, which is represented by the atmospheric transmissivity factor tau ($\tau$); the second term is the equivalent digital signal from the reflected radiation originating from the target object's surroundings ($S_{refl}$), also attenuated by the atmosphere; and the last term is the equivalent digital signal originated from the atmosphere itself in the path between the object and the sensor ($S_{atm}$).

### Estimation of atmospheric transmissivity

There are many different models available to estimate atmospheric transmissivity. For short distances, simple models that take into account the amount of water in the air can provide adequate estimates. For long distances (*e.g.*, for infrared cameras used in satellites), more sophisticated models which take into account not only water but also carbon dioxide, ozone, and other molecules, and other atmospheric factors such as scattering are used (*Gaussorgues, 1994*; *Zhang et al., 2016*). In this article, the method used in FLIR Systems' cameras was adopted (*FLIR Systems, 2001*), which estimates atmospheric transmissivity($\tau$) based on air water content ($H$), calculated from air temperature ($t$) and relative humidity ($RH$), and the distance between the object and the sensor ($d$):

$$H = RH \cdot e^{(1.5587 + 6.939 \cdot 10^{-2} \cdot t - 2.7816 \cdot 10^{-4} \cdot t^2 + 6.8455 \cdot 10^{-7} \cdot t^3)}, \tag{6}$$

$$\tau = X \cdot e^{[-\sqrt{d} \cdot (\alpha_1 + \beta_1 \cdot \sqrt{H})]} + (1 - X) \cdot e^{[-\sqrt{d} \cdot (\alpha_2 + \beta_2 \cdot \sqrt{H})]}. \tag{7}$$

### Estimation of digital signal values for different radiation sources

Assuming all temperatures and emissivities are known, the signal values originating from the different radiation sources, which contribute to the total signal produced by the sensor, can be estimated using Eq. (4). For the target object, the signal ($S_{obj}$) can be calculated

based on the object temperature ($T_{obj}$) and its emissivity ($\varepsilon$):

$$S_{obj} = G \cdot \varepsilon \cdot \frac{1}{R \cdot (e^{\frac{B}{T_{obj}}} - 1)} + O. \tag{8}$$

The signal originated from the atmosphere between the object and the sensor ($S_{atm}$) can be calculated based on air temperature ($T_{atm}$) and its emissivity, which is equal to $1 - \tau$:

$$S_{atm} = G \cdot (1 - \tau) \cdot \frac{1}{R \cdot (e^{\frac{B}{T_{atm}}} - 1)} + O. \tag{9}$$

For estimating the signal from radiation reflected by the target object ($S_{refl}$), one must take into account the reflectivity of the object, which is equal to $1 - \varepsilon$. Also, it should be necessary to know the temperature of the surrounding objects ($T_{refl}$) and their emissivity ($\varepsilon_{refl}$):

$$S_{refl} = G \cdot (1 - \varepsilon) \cdot (\varepsilon_{refl}) \cdot \frac{1}{R \cdot (e^{\frac{B}{T_{refl}}} - 1)} + O. \tag{10}$$

Since in most cases it would be difficult to determine the temperature and emissivity of all the surrounding objects, the usual procedure is to estimate an aparent reflected temperature ($T_{app.refl}$), by measuring the apparent temperature of a reflective material with $\varepsilon \approx 0$ (usually aluminium foil). Using this procedure, Eq. (10) would be replaced by:

$$S_{refl} = G \cdot (1 - \varepsilon) \cdot \frac{1}{R \cdot (e^{\frac{B}{T_{app.refl}}} - 1)} + O. \tag{11}$$

### Object temperature calculation

In order to calculate object temperature ($T_{obj}$), it is necessary to first obtain the signal originating from the object by solving Eq. (5) by $S_{obj}$, and usign the total signal $S$ and the results from Eqs. 7, 9, and 11:

$$S_{obj} = \frac{S}{\tau} - S_{refl} - \frac{S_{atm}}{\tau}. \tag{12}$$

Finally, by solving Eq. (8) by $T_{obj}$ and using the result of Eq. (12) and the sensor's gain, offset and calibration parameters ($G$, $O$, $B$, and $R$), it is possible to calculate the object temperature as follows:

$$T_{obj} = \frac{B}{\log(\frac{G \cdot \varepsilon}{R \cdot (S_{obj} - O)} + 1)}. \tag{13}$$

## Implementation of the temperature calculation algorithm

IRimage was implemented in the macro language of the widely used, open source, scientific image analysis software ImageJ (*Rueden et al., 2017*) or its distribution FIJI (*Schindelin et al., 2012*), and also uses the open source software ExifTool (*Harvey, 2003*) to extract raw values from the thermal images. It was implemented and tested using FLIR brand cameras (FLIR Systems Inc., USA), which is one of the most widely used brands in research (*Harrap et al., 2018*).

**Table 1  Data and parameters extracted from FLIR radiometric JPEG images.**

| Parameter / Variable (Units) | Variable name in IRimage | Symbol used in equations | EXIF tag name in FLIR JPG file |
|---|---|---|---|
| *Sensor data* | | | |
| Raw sensor signal (DN) | rawSignal_DN | $S$ | Raw Thermal Image |
| *Calibration / camera-specific parameters* | | | |
| Raw Thermal Image Type (PNG or TIFF) | imageType | | Raw Thermal Image Type |
| Camera Model | cameraModel | | Camera Model |
| Sensor gain | sensorG | $G$ | Planck R1 |
| Sensor offset | sensorO | $O$ | Planck O |
| Sensor calibration parameter B | sensorB | $B$ | Planck B |
| Sensor calibration parameter F * | sensorF | | Planck F |
| Sensor calibration parameter R | sensorR | $R$ | Planck R2 |
| *Atmospheric parameters* | | | |
| Atmospheric transmissivity parameter 1 | atmAlpha1 | $\alpha_1$ | Atmospheric Trans Alpha 1 |
| Atmospheric transmissivity parameter 2 | atmAlpha2 | $\alpha_2$ | Atmospheric Trans Alpha 2 |
| Atmospheric transmissivity parameter 1 | atmBeta1 | $\beta_1$ | Atmospheric Trans Beta 1 |
| Atmospheric transmissivity parameter 2 | atmBeta2 | $\beta_2$ | Atmospheric Trans Beta 2 |
| Atmospheric transmissivity parameter X | atmX | $X$ | Atmospheric Trans X |
| *User-selected parameters* | | | |
| Apparent reflected temperature (° C) | appReflTemp_C | | Reflected Apparent Temperature |
| Air temperature (° C) | airTemp_C | $t$ | Atmospheric Temperature |
| Object emissivity | objEmissivity | $\varepsilon$ | Emissivity |
| Air relative humidity | airRelHumidity_perc | $RH$ | Relative Humidity |
| Object distance from camera | objDistance_m | $d$ | Object Distance |

**Notes.**

\*This parameter is included in the JPG EXIF tags but it is (usually) equal to 1, and is equivalent to the value of 1 in the term ($e^{\frac{B}{T}} - 1$) in Eq. (4).

### Extraction of parameters from JPEG files.

The temperature calculation method relies on having access to the raw sensor data obtained by the camera. In the case of FLIR cameras, this data is stored in the radiometric JPG files (the standard file format for these cameras) as metadata tags in ''EXIF'' format. This metadata also includes camera-specific and user-set parameters which are also used to calculate temperature. All the parameters that are extracted from the JPG files, and the corresponding variables used in IRimage are detailed in Table 1. First, IRimage uses the ExifTool software (*Harvey, 2003*) to processes all images in JPG format within the user-selected folder in order to extract the raw sensor data, which is stored as a PNG image. Next, all camera-specific, atmospheric and user-set parameters are extracted.

### Calculation of derived variables

The next step is the calculation of variables derived from these parameters (detailed in Table 2), including the calculation of atmospheric transmissivity (using Eqs. (6)–(7)) and the estimated signal from reflected objects and the atmosphere (using Eqs. (9)–(11)). The byte order (endianness) of the raw image is determined from the image type (PNG or TIFF). This works in almost all cases, but it has been found that this rule does not hold for

**Table 2  Variables derived from parameters.**

| Parameter/Variable (Units) | Variable name in IRimage | Symbol used in equations |
| --- | --- | --- |
| Raw image byte order/endianness | `byteOrderLittleEndian` | |
| Aparent reflected temperature (K) | `appReflTemp_K` | $T_{app.refl}$ |
| Air temperature (K) | `airTemp_K` | $T_{atm}$ |
| Air water content | `airWaterContent` | $H$ |
| Atmospheric transmissivity | `atmTau` | $\tau$ |
| Raw signal from atmosphere (DN) | `atmRawSignal_DN` | $S_{atm}$ |
| Raw signal from reflected radiation (DN) | `reflRawSignal_DN` | $S_{refl}$ |

**Table 3  Variables used for temperature calculation.**

| Parameter/Variable (Units | Variable name in IRimage | Symbol used in equations |
| --- | --- | --- |
| Raw sensor signal (DN) | `rawSignal_DN` | $S$ |
| Raw signal from object (DN) | `objRawSignal_DN` | $S_{obj}$ |
| Object temperature (°C) | `objTemp_C` | $T_{obj}$ |

some (at least three) camera models. In those cases, an exception to this rule is included in the code.

Depending on the option selected by the user, both the extraction of parameters and the calculation of these variables are either performed for each file or only for the first file in the folder (when the user wants to apply the same set of parameters to all files). In the latter case, the user can also modify the "user-selected" parameters.

### *Temperature calculation*

The variables used for the final temperature calculation are detailed in Table 3. The PNG image containing the raw sensor data is opened, and each pixel containing the digital signal from the sensor is processed sequentially. First, the object signal is estimated using Eq. (12), and then the temperature value is calculated using Eq. (13).

## Software usage

IRimage is run as a plugin of the scientific image analysis software ImageJ (*Rueden et al., 2017*) or its FIJI distribution (*Schindelin et al., 2012*), and also uses the open-source software ExifTool (*Harvey, 2003*) to extract raw values from the thermal images. IRimage is available for download at https://github.com/gpereyrairujo/IRimage. When IRimage is installed, an "IRimage" sub-menu is added to the "Plugins" menu of ImageJ/FIJI, which allows the user to access the available functions (Fig. 1). The four functions included in IRimage are: (1) "Process", which processes the original thermal images to extract raw data and estimate temperature; (2) "Measure", which allows the user to measure the temperature of different objects in complete sets of images; (3) "Color", to create false-color images and videos for reporting; and (4) "Test", to compare IRimage results against other software.

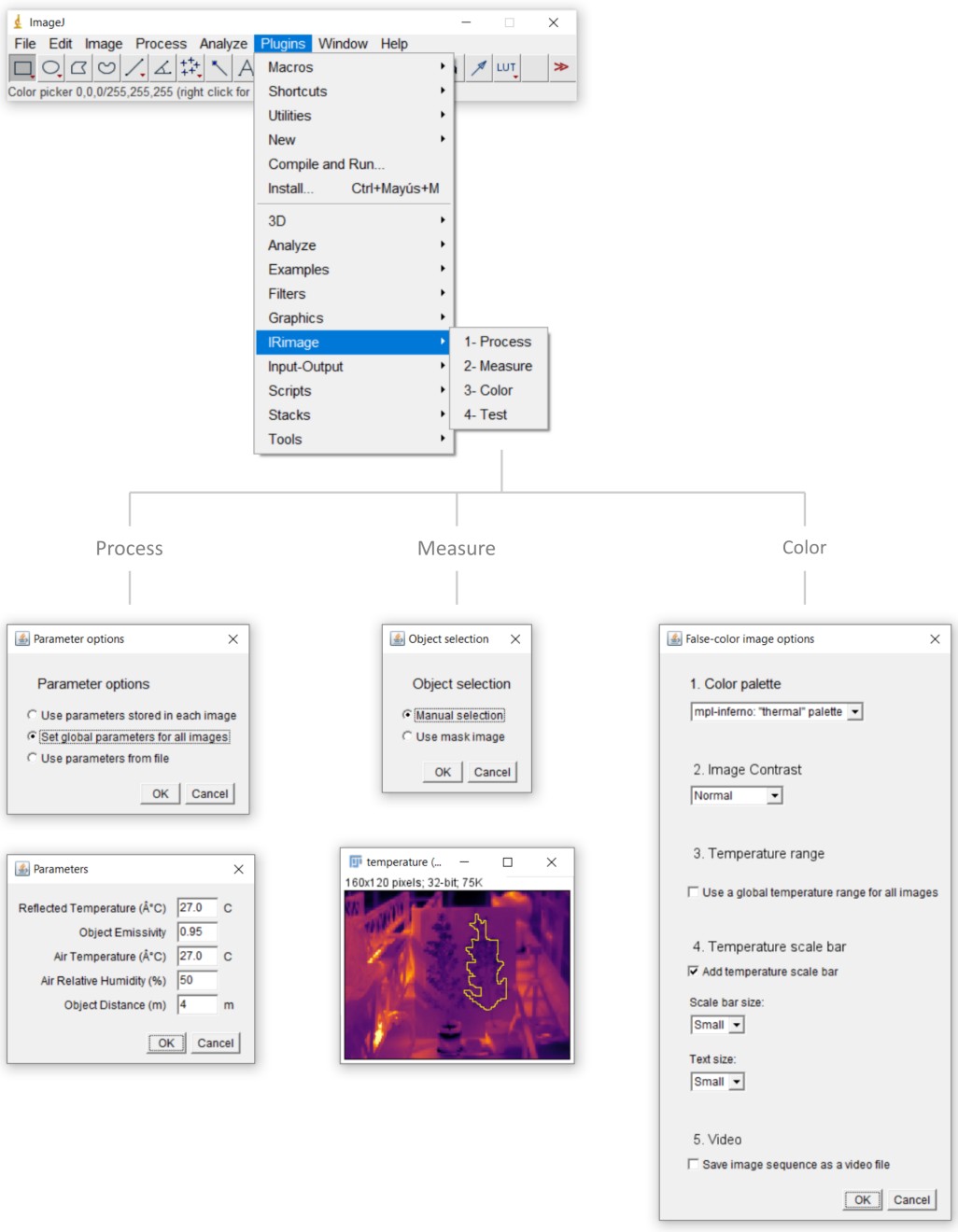

**Figure 1  IRimage's functions.** The ImageJ/FIJI "Plugins" menu containing the "IRimage" sub-menu and functions, and the main windows and dialog boxes that are shown to the user. In any of the functions, the user is first promoted to select the folder containing the original thermal images to be processed (not shown). In the "Process" function, the user is first asked to indicate how parameters for temperature calculation will be determined for each image and, if the user chooses to enter a set of values manually, a second dialog box is shown for the user to do so. In the "Measure" function, the user is first asked to choose whether to manually select the areas to be measured, or to use a previously saved mask. 

**Figure 1 (...continued)**
If the first option is selected, the user first enters the number of areas to measure (not shown) and then a window with the first image of the set is opened so that user can indicate the areas to be measured using any of the available selection tools. In the "Color" function the user can select the color palette, the contrast level, how to calculate the displayed temperature range, whether to include a temperature scale bar, and whether to produce a video file as output. The "Test" function does not require additional user input.

### Processing thermal images

IRimage was implemented and tested using FLIR brand cameras (FLIR Systems Inc., USA), which is one of the most widely used brands in research (*Harrap et al., 2018*), and is able to process FLIR's radiometric JPG thermal image format.

The "Process" function of IRimage processes complete folders of thermal images, since this is usually the case in research uses. The radiometric JPG image format includes user-set parameters (*i.e.,* emissivity, air temperature and humidity, reflected temperature, and object distance) within individual images, but it is also possible to use different parameter values to calculate temperature from raw data. IRimage can process the images using either of three options (Fig. 1): (1) process each file using the stored parameters (*e.g.*, when the user has manually set these parameters in the camera according to specific conditions for each image); (2) use a set of global parameters for all images, which is useful when all images were captured under the same conditions, or if parameters need to be modified globally (in this case a dialog box is shown where parameters can be set, Fig. 1); or (3) use parameters stored in a text file, in which specific parameters can be defined for each image.

IRimage processes all images in JPG format within the user-selected folder. Raw data containing the digital signal from the sensor is extracted and temperature is estimated for each pixel using the algorithm detailed in the "Theoretical background for temperature calculations" section. After processing, three images are stored for each input file, corresponding to the raw data, the estimated temperature, and a false-color image. The estimated temperature pixel values are also stored as text, in a .csv (comma-separated values) file that can be opened in a spreadsheet or statistical software. Also, irrespective of the processing option selected, the parameters used are stored in a .csv text file which can be later used to reproduce the same results (with the "Use parameters from file" option, Fig. 1). Each of these output file types are stored in different subfolders within the user-selected folder.

### Measuring the temperature of objects

A third function is included to perform reproducible measurements of the temperature of objects in the images. With the "Measure" function (Fig. 1), the user can select up to 255 different objects (rectangular, oval, or free-form areas, lines or points) and obtain temperature measurements (mean, minimum, maximum and standard deviation) for each of them, for each of the images in a folder. The selected objects are stored in a "mask" image, which can be later used to reproduce the same measurements. A mask image can also be modified or created using other methods and used in IRimage.

### Creating customized false-color images and videos

The "Color" function allows the user to produce false-color images representing the temperature values, which are useful for visualizing and reporting (Fig. 1).

The default palette used in IRimage is "mpl-inferno", one of the default palettes in the popular data visualization Python package Matplotlib (*Van der Walt & Smith, 2015*), which is also included in FIJI. It is a typical "thermal" palette that represents colder values in dark blue, and transitions through red and yellow for warmer values (emulating the color of hot liquid metal, or the radiation emitted by a black body). It was selected because it is both perceptually uniform and suitable for color-blind viewers. Alternatively, the classic greyscale palette, representing colder values as black and warmer values as white, is supported as well. Another important aspect is selecting the appropriate temperature scale, in order to efficiently represent the temperature values in the images using the full color palette. IRimage allows the user to select the contrast level, by automatically adjusting the minimum and maximum displayed values through a "histogram stretching" algorithm (*Fisher et al., 2003*). It operates by setting an amount of pixels with extreme values (the "tails" of the histogram) that are excluded (0, 0.3 and 3% for the low, normal and high contrast options, respectively). There are also two ways to calculate the temperature range for a set of images: either use the same scale for all of them (based on the range of temperatures in the full image set, which allows for a better comparison between images), or adjust the scale to the temperature range in each image (which allows for a better visualization of temperature differences within each image). It is also possible to add a scale bar to the images, showing the temperature scale and the color palette, with two different sizes for the scale bar and the font. Lastly, the user can choose whether to produce a video file for the complete set of images as a sequence.

### Testing the temperature estimations against other software

The Test function provides a way of testing the algorithm, by comparing the results obtained with the Process function of IRimage against data exported using another software (*e.g.*, that provided by the camera manufacturer). Since IRimage is an open-source software and therefore its modification and customization is possible (and encouraged), this function can be used to check if the calculations have not been altered by any change in the code made by the user (for that purpose, a test image is included with IRimage, along with the temperature data exported using FLIR Tools). Also, it can be used to check whether IRimage functions correctly for a given camera when it is used for the first time. After the function is run, a scatter plot is drawn for each image for which there is reference data available, comparing temperature values of all the pixels, and a text file indicating the mean and maximum temperature differences.

## Comparison to existing tools

IRimage was evaluated by comparing the resulting temperature values with those exported manually using the FLIR Tools software (FLIR Systems, Inc., USA, version 5.13.18031.2002). A wide set of 26 images taken with 15 different camera models was used for this comparison. Images were downloaded from *Wikimedia Commons (2019)*, and are

listed in Table S1. All the images were first processed using IRimage using the user-defined parameters stored in the image file. After that, each file was opened using FLIR Tools and the temperature values were manually exported in csv format. Finally, the temperature values for each image were compared using the Test function in IRimage.

### Example use case

Two *Pittosporum tenuifolium* (Banks & Sol. ex Gaertn) plants, grown in soil-filled, 3.5L pots, were placed inside a greenhouse and imaged with an infrared thermal camera (FLIR E40bx, FLIR Systems, USA) every ≈6 min. during 24 h., from a distance of 4 m. The camera was triggered automatically by means of a custom-built device (*Pereyra Irujo, 2019*). Air temperature, relative humidity, and incident photosynthetically active radiation were measured every 1 min. with a datalogger (Decagon Em50, Decagon, Pullman, WA, USA). Plants were not watered during the previous 4 days, and before the onset of measurements, 0.5 L water was added to one of them, so that soil water content was raised from ≈0.2 to ≈0.4 $m^3/m^3$.

Images were captured with a fixed set of user-defined parameters (since capture was automated) but then processed using IRimage using unique values for each image. Reflected temperature was estimated for each image similarly to the usual procedure (*FLIR Systems, 2016*), as the mean temperature of a piece of aluminium foil placed in the camera's field of view measured in the images processed using an emissivity value of 1. Air temperature and relative humidity used were those measured by the weather sensor at the time each image was captured. These values were entered in a .csv text file which was then selected through the "Use parameters from file" option of the "Process" function.

After processing, the leaf temperature of each of the two plants was measured in all the images. Due to the intricate shape of the plants, a mask image containing the leaf pixels to be measured was created using ImageJ, using a combination of a thresholding algorithm and manual selection. For these measurements, the images were processed using an emissivity parameter of 0.95 (*Salisbury & Milton, 1988*), and a distance between the camera and the objects of 4 m.

Leaf temperature was used in combination with sensor data for air temperature to calculate the leaf-to-air temperature difference ($\Delta T$), a key variable for analyzing the energy balance of the plant (*Gates, 1964*). The temperature of the air sensor enclosure (also placed in the camera's field of view) was measured in the images as well, using a similar approach and an emissivity value of 0.84 (*FLIR Systems, 2016*).

## RESULTS

### Comparison to existing tools

Figure 2 shows a sample of the images used, either in their original form (Figs. 2A–2C) or processed with IRimage (Figs. 2D–2F), and scatter plots comparing the temperature values for all pixels obtained with IRimage *vs.* FLIR Tools, either showing the full range of temperatures in the image (Figs. 2G–2I) or a "zoomed-in" version showing a small range of temperatures in detail (Figs. 2J–2L). The comparison between IRimage and the manufacturer's software showed an average difference of 0.0002 °C, and in all cases below

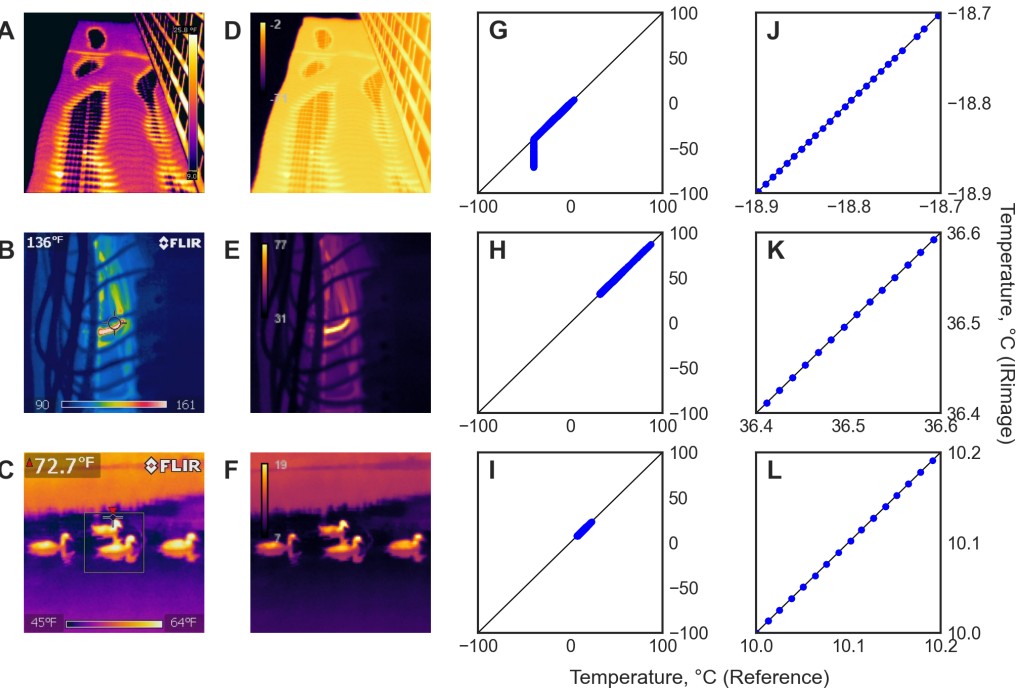

**Figure 2** **Results of the comparison between IRimage and FLIR Tools in a subset of the 26 images used.** (A–C) Original images used. (D–F) Images produced by IRimage with its perceptually uniform color palette. (G–I) Scatter plots comparing the temperature values of all the pixels in each image obtained using IRimage *vs.* the reference temperature values obtained from the original images using the manufacturer's software. (J-L) Scatter plots showing in detail a subset of the same data around the mean temperature of the image ±0.1 °C. Original images downloaded from *Wikimedia Commons (2019)*: (A) https://commons.wikimedia.org/wiki/File:Aqua_Tower_thermal_imaging.jpg (CC0 license), (B) https://commons.wikimedia.org/wiki/File:200_deg_neutral.jpg (CC0 license), (C) https://commons.wikimedia.org/wiki/File:Thermal_image_of_four_ducks_swimming.jpg (CC0 license).

0.01 °C, but only when temperature was above −40 °C. When temperature was below that value, temperature obtained using FLIR Tools was always equal to −40 °C, irrespective of the initial raw values, whereas IRimage showed values that could reach −70 °C, as can be noted in Fig. 2I. Out of the 26 images analyzed, in only one case the comparison was not possible because the temperature data could not be extracted using FLIR Tools (although it was possible to process it with IRimage). The comparison between IRimage and FLIR Tools for the 25 images that could be analyzed is presented in Fig. S1.

## Example use case

Air temperature within the greenhouse ranged from 20 °C early in the morning to 43 °C at around noon, and air relative humidity from 13% during the day to 78% in the night (Fig. 3). Reflected temperature (measured in the images as indicated previously) ranged from 21 to 46 °C. Incident solar radiation reached 1,868 $\mu$mol m$^{-2}$ s$^{-1}$, totalling 25.5 mol m$^{-2}$ d$^{-1}$. Measuring the temperature of the sensor enclosure using the thermal images yielded temperature values similar to those measured with the sensor, as shown in Fig. 3B.

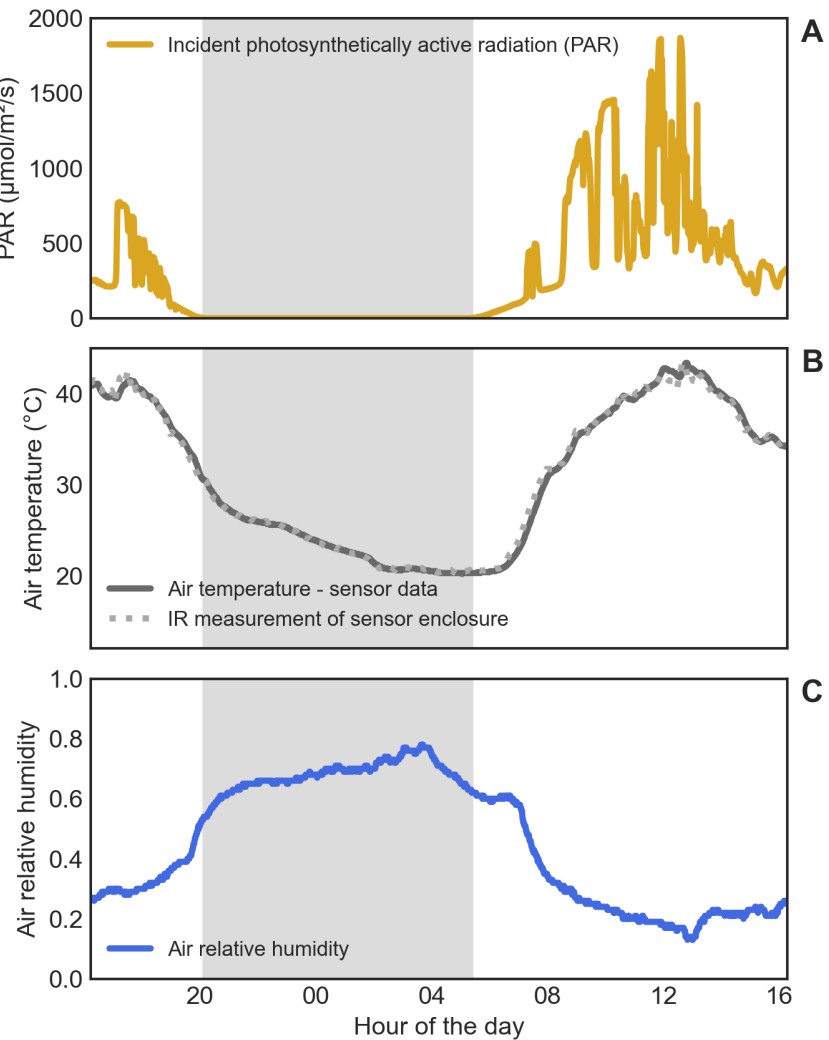

**Figure 3** **Environmental conditions in the greenhouse during the experiment.** (A) Incident photosynthetically active radiation (PAR). (B) Air temperature, measured either with the weather sensor (continuous line) or by measuring the sensor enclosure with the thermal camera (dotted line). (C) Air relative humidity. The gray shaded area indicates the night.

Leaf temperature, as measured with the thermal camera, seemed to follow air temperature closely, as shown in Fig. 4A. Nevertheless, $\Delta T$ curves revealed temperature differences between leaves and air, especially during the day (Fig. 4B). A heating effect of solar radiation incident on the leaves could be seen early in the morning (reaching 2 °C above air temperature), followed by a cooling effect of transpiration in the following hours (reaching 1 to 2 °C below air temperature), with fluctuations that follow the changes in the amount of incident solar radiation. Plants also showed differences in leaf temperature between them, but mainly during the day, indicating a restricted transpiration (and therefore less evaporative cooling) in the water-stressed plant (Fig. 4B). A Supplementary Figure (Fig. S2) is included showing the difference between these results and those that are obtained using

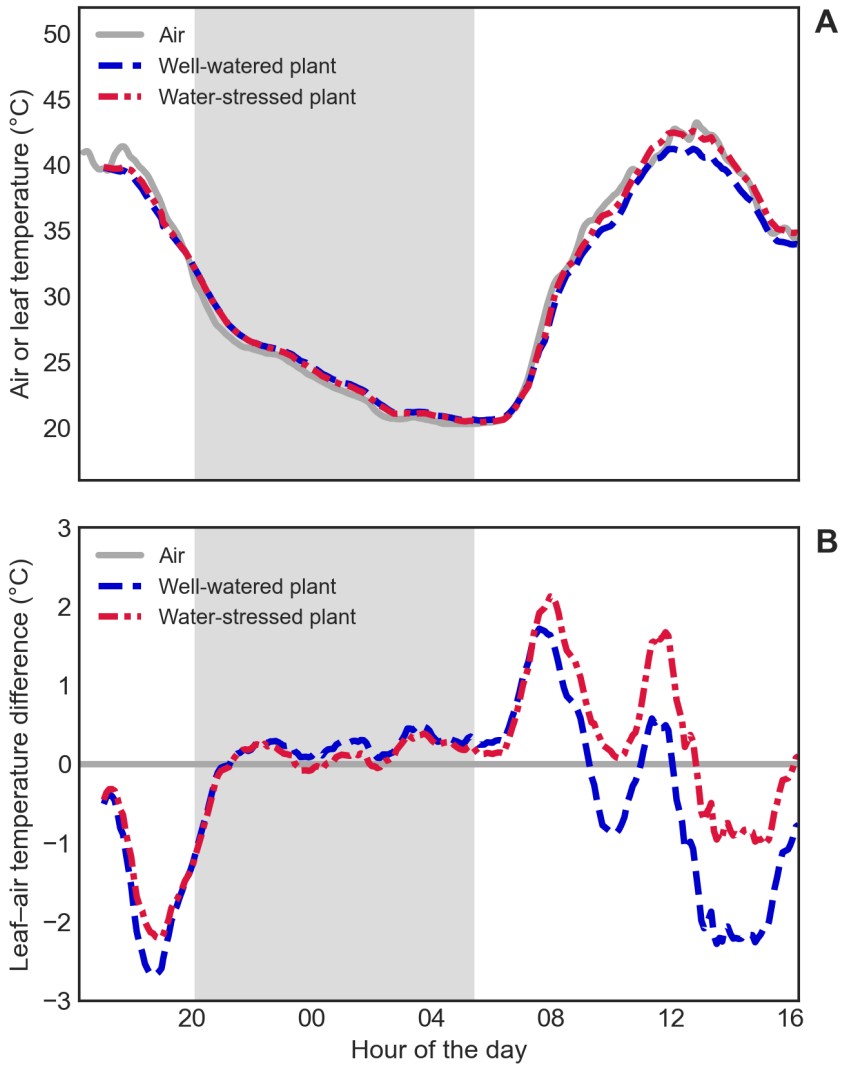

**Figure 4** **Leaf temperature of well-watered and water-stressed plants.** (A) Absolute air temperature (grey solid line) and absolute leaf temperature of the well-watered (blue dashed line) and water-stressed (red dashed-dotted line) plants. (B) Leaf-to-air temperature difference ($\Delta T$) of the well-watered (blue dashed line) and water-stressed (red dashed-dotted line) plants. The gray shaded area indicates the night.

fixed values for user-defined parameters (the normal procedure with standard software), that yield leaf temperatures values almost 1 °C higher or lower when environmental conditions deviate from average values.

Figure 5 shows a small sample of the 247 thermal images captured during the experiment, from selected moments of the day. In Fig. 5A, the color palette represents the same temperature range in all three images, which aids in visualizing the differences in absolute temperature between the night (00 h), the early morning (08 h), and the afternoon (14 h). Figure 5B shows the same images, but the color palette was set according to the temperature range in each image, which makes the comparison between them more difficult (or even misleading), but allows for a better perception of temperature differences within each

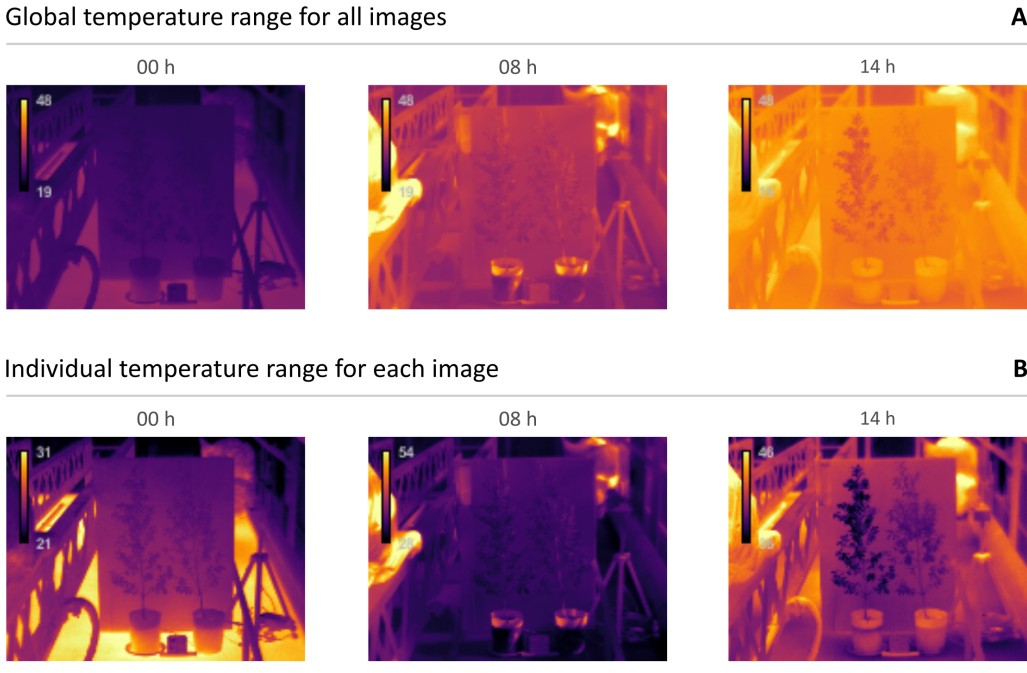

**Figure 5** **Example thermal images of well-watered (left) and water-stressed (right) plants in different moments of the day.** (A) Images taken in the night (00 h), the early morning (08 h), and the afternoon (14 h), and shown using a temperature scale (19–48 °C) set according to the global temperature range of the full set of images. (B) The same three images, shown using individually set temperature scales, according to the temperature range in each of them.

image. For instance, the image taken in the afternoon (at 14 h) shows the moment in which leaf temperature differs the most between the well-watered (in the left) and water-stressed (in the right) plants; that difference is much more discernible in the image shown in Fig. 5B. Two videos are included as supplementary material, showing the complete sequence of 247 images, either using a fixed scale according to the full temperature range (Video S1) or a variable scale according to the individual temperature range in each image (Video S2).

## DISCUSSION

Open source software is ideal for scientific research because it can be freely inspected, modified, and redistributed (*Schindelin et al., 2015*). IRimage is itself open source, and it was also implemented as a plugin of the open source software ImageJ, a widely used scientific image analysis tool which has been considered among the top "computer codes that transformed science" (*Perkel, 2021*). This not only provides a clear way of knowing the exact steps taken to estimate temperature from raw sensor data, but also allows the researcher to either use the tool through ImageJ's graphical user interface (without requiring any programming knowledge) or to modify, adapt or expand the functionality of the tool using ImageJ's powerful scripting languages (*Cacciabue, Currá & Gismondi, 2019*). ImageJ (especially its FIJI distribution) provides a large ecosystem of tools with which IRimage can interact, for example, by assembling different processing steps into pipelines

using scripting languages: plugins for image transformation, registration, annotation, enhancement, segmentation, visualization, as well as tools for interoperability with other software (*Schindelin et al., 2012*). Also, IRimage has been developed using the simple ImageJ macro language with the explicit aim of encouraging users to modify and contribute to improving the software.

IRimage provides tools for extracting and analyzing temperature data which are compatible with reproducible image handling recommendations (*Miura & Nørrelykke, 2021*), allowing researchers to avoid difficulties which are common when dealing with thermal images obtained with consumer-grade cameras. Software provided with these cameras has functions usually limited to temperature measurements of manually selected points or areas in the image (*e.g.*, *García-Tejero et al., 2018*; *Nosrati et al., 2020*). When the number of images is large, researchers resort to building custom *ad hoc* software, which is frequently not available for other researchers to reuse (*e.g.*, *Razani, Parkhimchyk & Tabatabaei, 2018*; *Van Doremalen et al., 2019*; *Goel et al., 2020*; *Mul Fedele et al., 2020*) or using elaborate methods to extract temperature values from color data in false-color images (*e.g.*, *Alpar & Krejcar, 2017*; *Petrie et al., 2019*). Moreover, in some cases these custom methods use "optimized" images produced by many consumer-grade cameras, which are meant for visualization and are a result of blending visible and thermal images for improved resolution, making the resulting data prone to errors.

One important aspect to consider when reporting temperature data as thermal images is choosing the right color palette (*Crameri, Shephard & Heron, 2020*). Some color palettes used for scientific visualization, such as the popular "rainbow" palettes, have a non-linear or unintuitive relationship between intensity and the represented value, and are not suitable for color-blind people (*Thyng et al., 2016*; *Nuñez, Anderton & Renslow, 2018*). Another problem of non-linear palettes is that they are not suitable for conversion to black and white, which can be important in journals that publish images in color online but not in its print version, or when the reader prints a journal article in black and white. The default palette ("mlp-inferno") used in IRimage was chosen so as to avoid those pitfalls, and therefore aid in accurate data interpretation and communication. Other similar palettes exist, also suitable for scientifically representing thermal data, such as "mpl-magma", "mpl-plasma" (*Van der Walt & Smith, 2015*), "CET L-03", "CET L-08" (*Kovesi, 2015*), or "cmocean-thermal" (*Thyng et al., 2016*).

The example use case presented in this paper highlights the utility of some of IRimage's functions, obtaining data that revealed subtle differences in the daily pattern of leaf temperature of two plants that had received more or less irrigation water. Obtaining this data required analyzing 247 thermal images, each of them with different parameters for temperature estimation. This task would have been extremely impractical with standard software, in which each image has to be manually processed. Also, measurement of leaf temperature of the irregular-shaped plants should have probably had to be simplified to square or oval shapes, and these areas selected manually for each image, which would have increased measurement error (by including background pixels) and prevented later reproduction of those results (if the selected areas are not repeated exactly). Also, the presented example highlights how the usual procedure of setting camera parameters (air

temperature, relative humidity and reflected temperature) before capturing a large set of images (especially in fluctuating environmental conditions) can lead to potentially large measurement errors, even when these parameters represent the true average conditions. IRimage allows to efficiently and reproducibly process images using unique parameters for each image, thus obtaining more accurate results.

One of the main drawbacks of IRimage is that it is now limited to only one brand of thermal cameras, since it has been implemented and tested for FLIR brand cameras. Its utility is, however, large enough, since this brand is, by far, the most widely used in biological research: in a systematic literature review of thermography in biology, 61% of papers reported using FLIR cameras (FLIR Systems Inc., USA), followed by NEC (NEC Ltd., Japan) and Fluke (Fluke Corporation, USA), with 7% each, and InfraTec (InfraTec GmbH, Germany) with 4% (*Harrap et al., 2018*). The algorithms are, nevertheless, potentially adaptable for other cameras for which raw sensor data could be obtained. Among the cameras from this brand, IRimage was able to process all the tested models (15), yielding individual pixel values that did not differ from those obtained with the manufacturer's software by more than 0.01 °C, and being able to process raw data that was below of the manufacturer's software limit of −40 °C. It should be noted that this value only represents the minimal discrepancy between different processing methods, but not the actual measurement error. Measurement errors can arise due to the hardware itself (the specified sensitivity of a thermal camera can be around 0.07 °C, with an accuracy of ±2 °C, *e.g.*, *FLIR Systems, 2016*) or the measurement technique, and so a periodic factory calibration or testing against a target of known temperature is advised (*e.g.*, *Klaessens, Veen & Verdaasdonk, 2017*).

## CONCLUSIONS

Affordable infrared thermal cameras have proven to be suitable for research, especially in low-resource settings, but the use of closed-source, consumer-oriented or custom software for image analysis can limit the throughput, accuracy and reproducibility of the results. IRimage provides open-source, flexible and documented tools for processing, measurement and reporting of thermal imaging data for research purposes in biological and environmental sciences. This tool includes functionalities that make it better suited for research purposes than many currently available alternatives, and could contribute to making affordable consumer-grade thermal cameras useful for reproducible research.

### Funding

This work was supported by Instituto Nacional de Tecnología Agropecuaria (PNCYO-1124072, 2019-PD-E3-I060), Universidad Nacional de Mar del Plata (AGR572/19, AGR637/21) and Agencia Nacional de Promoción Científica y Técnológica (PICT 2010-0006). The funders had no role in study design, data collection and analysis, decision to publish, or preparation of the manuscript.

## Grant Disclosures

The following grant information was disclosed by the author:
Instituto Nacional de Tecnología Agropecuaria: PNCYO-1124072, 2019-PD-E3-I060.
Universidad Nacional de Mar del Plata: AGR572/19, AGR637/21.
Agencia Nacional de Promoción Científica y Técnológica: PICT 2010-0006.

## Competing Interests

The author declares there are no competing interests.

## Author Contributions

- Gustavo Pereyra Irujo conceived and designed the experiments, performed the experiments, analyzed the data, performed the computation work, prepared figures and/or tables, authored or reviewed drafts of the paper, and approved the final draft.

## Data Availability

The software source code is available at GitHub: https://github.com/gpereyrairujo/IRimage.
The raw data, images, and code for data analysis is available at Github: https://github.com/gpereyrairujo/IRimage_paper.
Third party data was downloaded from Wikimedia Commons: https://commons.wikimedia.org.

## Supplemental Information

Supplemental information for this article can be found online at http://dx.doi.org/10.7717/peerj-cs.977#supplemental-information.

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
