# Peer review of "IRimage: open source software for processing images from infrared thermal cameras"

_PeerJ Computer Science, doi:10.7717/peerj-cs.977_

## Round 0.1 · original submission · Major Revisions

Reviews indicate the need for major revisions. Kindly provide a detailed point-to-point reply to reviewer comments when submitting a revised version.

Reviewer 1 ·

Basic reporting

This contribution is about processing of thermal images.
In general, this is a complex problem, users usually want to determine the temperature of the object, but thermal cameras measure the thermal radiation of the thin outer layer of the object.
The English is ok, but I am not native speaker, and I am not relevant to evaluate English.
Literature review is sufficient, the article structure is appropriate.
As I am first time as reviewer in this journal, it is a little bit difficult for me to find additional information.

Experimental design

The research should be supported by a physical overview. The reader appreciated that.
You write: the theoretical background of the algorithm used in IRimage and how it was implemented is included in this paper as Article S1
First time I downloaded only the main text.only and I didn’t find more information about physical overview. I don’t find in this paper Article S1 (I find it later), why it is divided on more parts? Maybe it's complicated from the publisher
This article (main text) is basic introduction to your system (software) with case projects, but it seems so inappropriate to me as a scientific paper for journal.
Only some sentences and reference to S1.are In material and methods, and so the article looks just like a guide to the software and usage examples.
However, this research is certainly interesting, usable, and valuable.

Validity of the findings

The article presents usable software for processing thermal images. Cheap thermal cameras only output to jpg and do not give raw data, this is an example of thermal cameras on a DJI Mavic drone, for example, which complicates scientific use.

Additional comments

General comments: this article summarizes the state of the art of using low-cost thermal cameras and gives an opportunity to process data from these instruments based on scientific research. Research and information are useful for practice. I recommend the article for publication. But I have a more fundamental note to the overall text: why is the theoretical part outside the article? I would like to at least welcome the extension to essential physical information and fundaments in the Materials and methods section.

Reviewer 2 ·

Basic reporting

This paper describes the functionality of the open source software IRimage which can be used for processing raw thermal images taken with several cameras of one brand FLIR and can potentially be used for other brand cameras. Some example results are presented on 14 thermal images obtained from a (non-verified) public source of thermal images ( Wikimedia ). In addition, an example use case of monitoring the temperature of plant leaves is presented of well and non-well hydrated plants during 24 cycles where many parameters vary in time.
The paper is clearly written in professional English language. The introduction give a good background in support of the need and usability for the software IRimage presented.

Experimental design

The author states that in this paper the software is validated against standard software. The reviewer does not agree that the method used can be considered as scientifically validation procedure using some sample images of a non-verified source and presenting the results in graphs at large scales (e.g. figure 2) were a difference of even 5 degree cannot be distinguished. In the example use case, too many variables have an influence on the measured/calculated temperature to prove the accuracy of the software. The details from the case presented distract from the purpose of this paper to present the IRimage software and could be a paper by itself.

For a true validation process, the thermal images should be taken with different cameras of the same object under controlled conditions. Not having all these camera available, the reviewer understand that this is a difficult task. However, with only 3 different cameras, the author would already present the validity of the software.

As example case, it would be more convincing to have only 1 or 2 variables for the temperature changes in time compared to a temperature reference e.g. a black body source temperature controlled by a current (temperature phantom). The case with the leaf temperature presented is nice as example for the many processing possibilities but cannot be claimed as a confirmation for accuracy.

Validity of the findings

If the author would adapt the text by stating some examples are presented of processing some random thermal images using IRimage it would be acceptable. Still the temperature comparison graphs like in figure 2 need to be presented zoomed in on the temperature scale of e.g. 10 degree so the accuracy within 0.1 degree can be distinguished.

Additional comments

The reviewer is convinced the software IRimage will be well received by the scientific community as a useful tool to process raw data from thermal cameras as long as this data is accessible. It would be great if this would also be used for small thermal imagers like the FLIR-One. This paper gives a good introduction in the potentials of the software. However, a scientific validation process is needed in which other researcher could contribute. The paper should be adapted not claiming the software has been validated by the examples presented. The example case could be presented with far less detail or replaced by a more simple case as suggested.

---

## Round 0.2 · Minor Revisions

The review indicates a minor revision. Authors are requested to make this revision and submit.

Reviewer 2 ·

Basic reporting

The reply of the author and revisions in the manuscript are adequate and meet the expectations of the reviewer. The 'claim' that the software is validated have been removed and replaced by 'compared to existing software'. It would still be preferred to have the software validated in a controlled setting with various thermal cameras (in the future paper). Now there is no claim of validation, the case presented is a nice illustration but still a more simple case would be preferred with less variables that can influence the temperature.

Experimental design

In this revision more theoretical background is given on the algorithm and parameters used to determine the object temperature. This greatly improves the scientific basis for the paper.
No additional comments compared to the first version

Validity of the findings

The presentation of the thermal images in figure 2 extended with a detailed section, as suggested by the reviewer, enables a better comparison (and convincing) between IRimage and FLIR Tools down to 0.1 degrees.
The example use case is presented in the results extensively and could be shortened e.g. leaving figure 6 out.

---

## Round 0.3 · accepted · Accept

The revisions are suitably addressed and accepted.

---

## Author Rebuttal · Round 0.3

REF: PeerJ Computer Science MS # 67901 - "IRimage: Open source software for processing images from infrared thermal cameras"

March 30th, 2022

Dr. Alex James

Academic Editor
Dear Editor,

I would like to thank again the reviewers and Editor for their comments.

I have edited the manuscript to address the request for minor revisions. Figure 6 and the associated text in the methods section (lines 262-275 in the MS with tracked changes) and results section (lines 301-305 and 332-337) were removed as requested, and now the description of the example use case is significantly shorter. The removed Figure is now included only as supplementary material. Also, an error in the species name of the plants used was corrected (line 254).

I believe that the manuscript is now suitable for publication in PeerJ Computer Science

Sincerely,

Dr. Gustavo Pereyra Irujo

INTA-CONICET